# Reduction in Five Harmful Substances in Fried Potato Chips by Pre-Soaking Treatment with Different Tea Extracts

**DOI:** 10.3390/foods12020321

**Published:** 2023-01-09

**Authors:** Weitao Wang, Huaixu Wang, Zhongjun Wu, Tingting Duan, Pengzhan Liu, Shiyi Ou, Hani El-Nezami, Jie Zheng

**Affiliations:** 1School of Biological Science, University of Hong Kong, Pok Fu Lam Road, Hong Kong 999077, China; 2Department of Food Science and Engineering, Jinan University, Guangzhou 510632, China; 3Guizhou Institute of Plant Protection, Guizhou Academy of Agricultural Sciences, Guiyang 550006, China; 4School of Food Science and Engineering, South China University of Technology, Guangzhou 510641, China; 5Guangdong-Hong Kong Joint Innovation Platform for the Safety of Bakery Products, Guangzhou 510632, China; 6Institute of Public Health and Clinical Nutrition, School of Medicine, University of Eastern Finland, FI-70211 Kuopio, Finland

**Keywords:** potato chips, tea extract, 5-hydroxymethylfurfural, acrylamide, dicarbonyl compounds, advanced glycation end-products

## Abstract

Thermally processed food always contains various types of harmful substances. Control of their levels in food is important for human health. This work used the extracts from green tea dust, old green tea, yellow tea, white tea, oolong tea, and black tea to simultaneously mitigate diverse harmful substances in fried potato chips. The six tea extracts (30 g/L) all showed considerable inhibitory effects on the formation of 5-hydroxymethylfurfural (reduced by 19.8%–53.2%), glyoxal (26.9%–36.6%), and methylglyoxal (16.1%–75.1%). Green tea and black tea extracts exhibited better inhibitory abilities than the other three teas and were further investigated for other harmful compounds by various concentration treatments. Finally, pre-soaking of fresh potato slices in 50 g/L extracts of green tea dust displayed, overall, the most promising inhibitory capacity of HMF (decreased by 73.3%), glyoxal (20.3%), methylglyoxal (69.7%), acrylamide (21.8%), and fluorescent AGEs (42.9%) in fried potato chips, while it exhibited the least impact on the color and texture. The high level of catechins in green tea dust may contribute most to its outstanding inhibitory effect, whereas the distinguished inhibitory effect of black tea extract was speculated to be attributable to the high levels of theaflavins and amino acids in the fully fermented tea. This study indicated that green tea dust, a predominant waste of the tea industry, had great potential to be exploited to improve food quality and safety.

## 1. Introduction

Thermal processing enhances the taste, aroma, color, and shelf life of foods. Nevertheless, it also produces various harmful substances, such as acrylamide, 5-hydroxymethylfurfural (HMF), α-dicarbonyl compounds, and advanced glycation end-products (AGEs), especially in foods rich in carbohydrates and lipids [1,2,3]. Acrylamide displays neural, genetic, and reproductive toxicities, and is categorized as a Group 2A carcinogen by the International Agency for Research on Cancer [4]. HMF is considered to be a dietary health risk mostly due to its mutagenic and carcinogenic metabolic product, 5-sulfooxymethylfurfural (5-SMF), which is formed through sulfonation in the body by sulfotransferases [5]. Glyoxal (GO) and methylglyoxal (MGO) represent typical α-dicarbonyl compounds, which are highly reactive to modify amino acids in proteins and protein residues through glycation reactions to form advanced glycation end-products (AGEs). The accumulation of AGEs in the body plays an important role in the development of various age-related chronic diseases, such as diabetes, cardiovascular disease, neurodegenerative disorder, and cancer [6]. Moreover, α-dicarbonyl compounds and 5-hydroxymethylfurfural also act as reactive precursors for the formation of acrylamide [3].

Fried potato chips are globally popular food as well as major dietary sources of the hazardous substances mentioned above. The acrylamide content (up to 4180 µg/kg) in most fried potato chips exceeds the European Commission’s recommended value of 750 µg/kg [7]. Exposure of potato tubers to UV-C radiation might even increase acrylamide content in the products [8]. High contents of HMF (up to 9.3 mg/kg), GO (up to 14640 µg/kg), and MGO (up to 6610 µg/kg) also feature in fried potato chips [9,10]. Thus, it becomes crucial and highly desirable to mitigate these harmful substances in fried potato chips. Sobol et al. [8] reported that soaking semi-products in water could wash out the reducing sugars, and resulted in a decrease in acrylamide content in French fries. In recent years, the utilization of additives, especially natural plant extracts rich in phenolic compounds, has become a popular and promising strategy to reduce the deleterious substances in food products [1,11,12,13].

Tea (*Camellia sinensis* L.) is a popular beverage with various health benefits related to its high levels of polyphenols and antioxidant activities. Its production reached 1.75 million tons in 2012 in China, and has been increasing continuously [14,15]. Tea is divided into six categories depending on different processing techniques: green tea (unfermented), yellow tea (slightly fermented), white tea (mildly fermented), oolong tea (semi-fermented), black tea (fully fermented), and dark tea (post-fermented) [16]. The differences in manufacturing processes dramatically determine the phenolic profiles and hence the antioxidant activities of the tea and its extract [17]. 

Utilization of tea extracts to inhibit hazardous substances in thermally processed foods is a promising strategy according to recent studies. Demirok and Kolsarici [18] reported that incorporation of green tea extracts lowered the level of acrylamide by up to 45% and 34%, respectively, in fried chicken drumsticks and chicken wings. Fu et al. [19] reported that the addition of (-)-epigallocatechin gallate extracted from green tea significantly reduced acrylamide in bread by 37%. The addition of green tea polyphenols decreased acrylamide in baked starchy matric by approximately 48%, while the reduction increased to 64% when green tea polyphenols was used in a combination with inulin [20]. However, the green tea extracts did not demonstrate an ability to mitigate acrylamide in rye bread [21]. This might indicate that the food matric determines the inhibitory capacity of the tea extracts on acrylamide, which needs to be investigated on a case-by-case basis. Other than acrylamide, the impact of tea extracts on the formation of other harmful substances was also investigated. A 55% reduction in HMF was detected in black garlic fermented by soaking in green tea extract [22]. Poojary et al. [23] reported that the green tea extract was capable of trapping HMF and dicarbonyl compounds, and inhibiting the formation of AGEs in UHT milk. However, most of the studies only focused on one or two of these harmful compounds. Actually, acrylamide, 5-hydroxymethylfurfural (HMF), α-dicarbonyl compounds, and AGEs are simultaneously produced in fried potato chips.

They share the same precursors and are produced from different intermediates. For example, HMF is formed from 3-deoxyosone, while glyoxal and methylglyoxal are formed from 1-deoxyosone, and share the same precursor―reducing sugar. It is possible that reducing one harmful compound will increase others.

Therefore, this study aimed to develop a method to decrease the generation of co-existing harmful substances simultaneously through the utilization of tea extracts, especially those derived from tea processing wastes. For this purpose, six different kinds of tea, namely, a green tea dust, an old green tea, a yellow tea, a white tea, an oolong tea, and a black tea, were compared for their inhibitory capacity against the formation of HMF, acrylamide, GO, MGO, and AGEs in fried potato chips by means of pre-soaking. Since different concentrations of tea extracts might result in distinct differences in the regulation of harmful compounds [24], the inhibitory effects of tea extracts of varying concentrations were further investigated to obtain the best mitigation strategy. The effects on texture and color were also evaluated for consideration in view of the sensory properties. The results of this study should provide practical strategies and useful guidelines for the control of harmful substances in thermally processed foods, and the improvement of food quality and safety. Meanwhile, through the utilization of byproducts and deposited tea waste, it also increases the added-value of teas and prevents the wasting of natural resources. 

## 2. Materials and Methods

### 2.1. Materials and Reagents

Raw potatoes, intended for production of fried potato chips in local food factories, were harvested in May 2022, and purchased from a local market. Tea samples, comprising a fresh green tea dust, an old green tea (stored for over a year), a black tea, and a white tea of *Camellia sinensis* L. cv. Fuding, and a yellow tea and an oolong tea of *Camellia sinensis* L. cv. Qianchayihao, were collected from the Tea Research Institute of Guizhou Academy of Agricultural Science, Guizhou, China. 2,2-Diphenyl-1-picrylhydrazyl (DPPH) radical scavenging assay kit and ferric-reducing antioxidant power (FRAP) assay kit were obtained from Suzhou Comin Biotechnology Co., Ltd (Suzhou, China) and Nanjing Jiancheng Bioengineering Institute (Nanjing, China), respectively. Edible soybean oil was purchased from Guizhou Sifang Cereals and Oils Co., Ltd (Guizhou, China). *o*-Phenylenediamine (OPD) was purchased from Shanghai Macklin Biochemical Co., Ltd (Shanghai, China). Folin–Ciocâlteu reagent was purchased from Hefei Bomei Biotechnology Co., Ltd (Hefei, China). 2,4-Dinitrophenylhydrazine was purchased from Tianjin Kermel Chemical Reagent Co., Ltd (Tianjin, China). Standards of catechin gallate (CG), epicatechin gallate (ECG), catechin hydrate (C), epicatechin (EC), gallocatechin (GC), epigallocatechin (EGC), gallocatechin gallate (GCG), and epigallocatechin gallate (EGCG) were purchased from Shanghai Yuanye Biotechnology Co., Ltd (Shanghai, China). Glyoxal (40% aqueous solution), methylglyoxal (40% aqueous solution), acrylamide, gallic acid, and 5-hydroxymethylfurfural were purchased from Shanghai Aladdin Biochemical Technology Co., Ltd (Shanghai, China).

### 2.2. Preparation of Tea Extracts

Tea samples were ground into fine powder before extraction. The sample of Fuding green tea dust was applied for extraction directly. The soaking solution was prepared by extracting 10–50 g (±0.001 g) of tea sample with 1 l of hot water (initially 100 °C) for 10 min under 200 rpm magnetic stirring, followed by another 30 min of ultrasonic extraction. The solution was vacuum filtered, and the supernatants were collected and cooled down.

### 2.3. Determination of Total Phenolic Content

The total phenolic content of each tea sample was determined according to the method applied by Fu et al. [25] with modification. Briefly, 1 mL of 10 g/L tea extract was taken, added to 5 mL 0.1 mol/L Folin–Ciocâlteu solution, followed by the addition of 7.5% (*w*/*v*) sodium carbonate solution to a final volume of 10 mL after 5 min. The mixture was incubated at room temperature for 1 h. Then, the absorbance of the mixture was measured at 765 nm. The total phenolic content of each tea sample was calculated by the calibration curve of the external standard of gallic acid.

### 2.4. Evaluation of Antioxidant Capacity

The antioxidant capacities of different tea samples were determined complying with manufacturer’s instructions. A total of 10 g/L tea extracts were used for the measurements, and the antioxidant capacities were finally calculated on the basis of the weight of tea samples. The radical scavenging capacity measured by DPPH kit was expressed as µmol Trolox/g, while FRAP value was described as mmol Fe^2+^/g.

### 2.5. Analysis of Catechins in Different Tea Samples

The tea extract (10 g/L) was diluted 1:50 in water and filtered through a 0.22 µm nylon syringe filter. The catechins in tea extracts were analyzed by the ultra-high-performance liquid chromatography-tandem mass spectrometry (UHPLC-MS/MS) system consisting of an UltiMate 3000 and a triple–quadrupole (QQQ) mass spectrometer equipped with an electrospray ionization (ESI) source (Thermo Fisher Scientific, Waltham, MA, USA). The compounds were separated on a Hypersil GOLD C18 column (100 mm × 2.1 mm i.d., particle size 1.9 µm; Thermo Scientific, Waltham, MA, USA). The mobile phases consisted of 0.1% formic acid in methanol (*v*/*v*; solvent A) and 0.1% formic acid in water (*v*/*v*; solvent B). The elution program was as follows: 0–1 min, 6.5% A; 1–2.5 min, 6.5%–32.5% A; 2.5–13 min, 32.5% A; 13–14 min, 32.5%–6.5% A; 14–15 min, 6.5% A. The flow rate was 0.2 mL/min, and the injection volume was 5 µL. The individual catechins were identified and quantified with the corresponding commercial standards by selected reaction monitoring (SRM) mode using an ESI mass spectrometer. The operating parameters of MS were as follows: capillary temperature, 300 °C; vaporizer temperature, 300 °C; spray voltage, 3500 V in the positive mode and 3000 V in the negative mode. The detection conditions and the fragmentation transitions for qualification and quantification are outlined in Appendix A.

### 2.6. Preparation of Potato Crips

The potatoes were washed, peeled, and cut into slices of 4.50 cm (±0.05 cm) in diameter and 0.10 cm (±0.01 cm) in thickness. Then, 100 g (±1 g) of potato chips was soaked in 400 mL of different tea extracts. The tea extracts were cooled to room temperature before use. Potato chips soaked in distilled water were applied as the blank group. After soaking for 60 min, the chips were taken out and drained. A household fryer (XML-EH81, Xuanhu Co., Ltd., Foshan, China) with a capacity of 6 l and equipped with a thermostat (±1 °C) was used for the frying process. Soybean oil (2 l) was loaded into the fryer and pre-heated to 160 °C. The chips were fried at 160 °C for 4 min, drained, and cooled to room temperature. The oil was replaced before each batch of chips was fried.

### 2.7. Color and Texture Analysis

Large and flat chip pieces were selected, and the color was measured using a TS7010 colorimeter (Shenzhen ThreeNH Technology Co., Ltd., Shenzhen, China) in five replicates on different slices from the same batch [26]. The sample was analyzed for color against a white background at room temperature, and the following component values were recorded: L*—brightness, chromaticity, a*—(from red to green), b*—(from yellow to blue) [27]. The texture was measured using a CT3 Texture Analyzer (Brookfield, NC, USA) equipped with a TA15 probe and TA-RT-KIT platform in seven replicates. The operating parameters of texture profile analysis (TPA) were set as: pretest speed, 2.00 mm/s; test speed, 1.00 mm/s; post-test speed, 1.0 mm/s; trigger force, 1 g; distance, 1.0 mm. As a rupture occurs during the first compression, a clear peak appears in the curve, and the force corresponding to this peak is defined as fracturability.

### 2.8. Water Extracts of Potato Chips

Water extracts of potato chips were prepared according to Huang et al. [26] with modifications. Briefly, 1 g (±0.001 g) sample of ground potato chips was de-fatted successively with 10 mL and 5 mL hexane. After the hexane residue was evaporated, 5 mL of distilled water was added to extract water-soluble components by ultrasonication (40 kHz) for 20 min and centrifuged at 5000 rpm for 20 min. Then, the extraction procedures were repeated twice with 2 mL of distilled water each. The supernatant from the three extractions was combined, fixed to a final volume of 10 mL, and filtered (0.22 µm). Three aliquots of water extracts were prepared for further analysis.

### 2.9. HMF Analysis

The method of HMF analysis was simply modified according to Yang et al. [28]. First, 1 mL of water extract was taken for the quantification of HMF using HPLC-UV 1260 (Agilent Technologies Co., Ltd., Santa Clara, CA, USA) with a Zorbax SB-Aq column (4.6 mm × 250 mm i.d., particle size 5 µm; Agilent Technologies Co., Ltd., Santa Clara, CA, USA) applied. A sample of 10 µL was injected and eluted at 0.8 mL/min with a constant mobile phase composed of water:methanol 95:5 (*v*/*v*). HMF was detected at 284 nm and quantified with the calibration curve of the external standard.

### 2.10. Acrylamide Analysis

The method for the determination of acrylamide was modified from Li et al. [29]. An UltiMate™ 3000 UPLC-Thermo TSQ QUANTUM ULTRA QQQ mass spectrometer (Thermo Fisher Scientific, Waltham, MA, USA) was applied for the analysis. First, 1 µL of the water extract was injected for separation on an ACQUITY UPLC BEC C18 column (2.1 mm × 50 mm i.d., particle size 5 µm, Waters, Milford, MA, USA) at 40 °C. The isocratic elution was performed using water:methanol 97:3 (*v*/*v*) as the mobile phase at a flow rate of 0.4 mL/min for 4 min. An optimized SRM method was conducted in positive mode for the detection and quantification of acrylamide. The instrument parameters were set as follows: capillary temperature, 220 °C; vaporizer temperature, 350 °C; spray voltage, 2500 V. A qualitative transition of 72.0 → 27.0 m/z (collision energy, 19 eV) and a quantitative transition of 72.0 → 55.0 m/z (collision energy, 9 eV) were applied, with a tube lens of 33 eV. Acrylamide was quantified with the calibration curve obtained with the commercial standard.

### 2.11. Determination of Glyoxal and Methylglyoxal

GO and MGO were determined according to the method by Huang et al. [26]. Briefly, 4 mL of water extract was reacted with 400 µL of 10% aqueous OPD (*w*/*v*) under pH 9.0 at 30 °C for 12 h in the dark. The mixture was filtered (0.22 µm) and analyzed with HPLC-UV 1260 (Agilent Technologies Co., Ltd., Santa Clara, CA, USA) using a Zorbax SB-Aq column (4.6 mm × 250 mm, 5 µm). The concentration of GO and MGO was determined with the calibration curves of the corresponding standard.

### 2.12. Evaluation of Fluorescent AGEs and Protein Oxidative Products

The fluorescent AGEs and three protein oxidation products (dityrosine, kynurenine, and *N*’-formylkynurenine) in water extracts of potato chips were measured as described by Huang et al. [26]. In brief, 300 µL of water extract from each sample was loaded into a black 96-well plate. The fluorescence intensity was measured using an Infinite M200 PRO multimode reader (Tecan Trading AG, Männedorf, Switzerland).

### 2.13. Determination of Carbonyl Value

The carbonyl value (CV) of the oil extracted from potato chips was determined according to Ou et al. [30] with modification. First, 5 g (±0.001 g) of ground potato chips was extracted by Soxhlet extraction with petroleum ether (200 mL) under reflux for 4 h. After extraction, the solvent was removed by a rotary vacuum evaporator at 40 °C, and the oil was collected. Then, 0.125 g (±0.001 g) of oil extract was weighed in a 25 mL stoppered test tube and dissolved with 5 mL of benzene. After 3 mL of trichloroacetic acid and 5 mL of 2,4-dinitrophenylhydrazine were added, the mixture was carefully shaken and reacted at 60 °C in a water bath for 30 min. The sample was cooled to room temperature with running water, and 10 mL of potassium hydroxide–ethanol solution was added, mixed by vortex shaking, and stood for 10 min. The absorbance was then measured at 440 nm.

### 2.14. Statistical Analysis

Data were expressed as mean ± standard deviation. The color measurements were performed in five replications, and the texture measurements were performed in seven replications. All of the other experiments were performed in triplicates. Shapiro–Wilk test was conducted to test the normality of the distribution. As a result, the *p*-values calculated by the Shapiro–Wilk test were all above the set level of significance (*p* > 0.05), which indicated that all the data follow a normally distributed population. The statistical differences among the samples were investigated by a one-way analysis of variance (ANOVA) at a confidence level of 0.05. Duncan’s multiple-range test for the population with equal variances and Tamhane’s test for that with unequal variances were employed to carry out the multiple comparisons at *p* < 0.05. Pearson’s correlation coefficients analysis was conducted to study the correlations between the total catechin contents and antioxidant capacities of tea extracts. All the statistical analyses were performed using SPSS Statistic 26 and Origin 2021.

## 3. Results and Discussion

### 3.1. Total Phenolic Content of Different Tea Samples

As shown in Figure 1A, the total phenolic contents were significantly different between the six kinds of tea. The fresh green tea dust had the highest value of total phenolic content (141.37 mg/g), followed by yellow tea (133.19 mg/g), white tea (119.51 mg/g), one-year-stored green tea (108.11 mg/g), oolong tea (97.59 mg/g), and black tea (32.23 mg/g). In the current study, green tea dust, white tea, and black tea were all freshly produced with an identical cultivar grown in the same location and year. The differences in the composition of these samples should be, therefore, mainly caused by the processing methods applied. Green tea is produced without fermentation. It undergoes a steamed and/or pan-fried process to inactivate the endogenous polyphenol oxidase (PPO) and hence minimize the oxidation, which potentially guarantees the maximum preservation of the phenolic compounds in the leaves [15]. White tea is a slightly fermented tea, the withering process of which was different from all the other kinds of Chinese tea. In comparison, black tea was fully fermented, where the leaves were rolled to disrupt cellular compartmentation and allow the contact of phenolic compounds with the polyphenol oxidases. The leaves undergo oxidation for 90−120 min before the drying process [31]. As a result, the amount of total phenolic compounds was comparably low in black tea in comparison to the other kinds of tea. The yellow tea and oolong tea in this study are of the same cultivar, Qianchayihao. The degree of fermentation of yellow tea was lower than that of oolong tea. Accordingly, yellow tea was more abundant in phenolic compounds than oolong tea (Figure 1A). Regardless of the cultivars, the tea samples almost showed a decreasing trend in total phenolic content as the fermentation degree of tea increased. Moreover, the significant decrease in total phenolic content in one-year-stored green tea indicated that prolonged storage would cause a remarkable destruction of and decrease in phenolic compounds.

### 3.2. Catechin Profile of the Tea Samples

Catechins are the predominant phenolic compounds in tea, and their profile is considered to be closely related to the bioactive profiles [32]. In this study, a total of eight catechins were identified and quantified in different tea samples (Table 1). The highest total content of catechins was detected in yellow tea (125.18 mg/g), followed by green tea (97.33 mg/g), green tea dust (86.09 mg/g), oolong tea (60.05 mg/g), white tea (50.75 mg/g), and black tea (0.58 mg/g). Jiang et al. [16] collected different types of teas from various geographical regions and determined the total catechin content in green tea, yellow tea, white tea, oolong tea, and black tea in the ranges of 60.33–145.62 mg/g, 34.3–92.58 mg/g, 39.75–67.31 mg/g, 40.65–97.79 mg/g, and 7.69–23.06 mg/g, respectively. The noteworthy differences in the catechin concentrations between the samples, as well as from our findings, can be attributed to the variation in cultivars and growth regions, which may also explain the significantly high levels of catechins (142–167 mg/g in total) reported in a ‘Huang Zhi Xiang’ oolong tea collected in Guangdong province of China [33]. Other studies reported similar total catechin contents to those detected in our study, varying from 92 to 119 mg/g in green teas [34,35]. Both our study and that of Jiang et al. demonstrated that black tea contained the lowest amount of catechins compared to the other types of tea. The endogenous enzymes (polyphenol oxidase and peroxidase) in tea leaves oxidize catechins during the typical fermentation process in black tea production, which results in a sharp decrease in the total catechin content. The enzymatic oxidation of the catechins yields certain water-soluble pigments, such as theaflavins, thearubigins, and theabrownin, which contribute greatly to the quality of black tea [36]. 

Among the eight catechins detected, ECG, EGC, and EGCG were the most abundant in the tea samples, but their proportion varied between different tea samples. EGCG was the most predominant catechin in green tea dust (56.43% of total catechins), yellow tea (36.66%), and white tea (43.90%). This was consistent with the findings of other researchers [34,35]. Given that the green tea dust, white tea, and black tea were freshly produced from the same batch of tea leaves, EGCG content displayed a clear decreasing trend as the fermentation degree increased. Accordingly, the EGCG content in the mildly fermented yellow tea was about three-fold higher than that in the semi-fermented oolong tea from the same tea batch. The other two major catechins, ECG and EGC, also showed decreasing trends in contents as the fermentation degree increased. Notably, the one-year-stored green tea contained four times higher EGC and EC than the fresh green tea dust. The contents of EGCG and ECG in stored green tea, however, were 30.12% and 64.65% lower, respectively, than those in the fresh green tea dust. Catechins easily undergo epimerization and degradation during thermal processing and storage [37]. However, since the two green tea samples were harvested and produced in different years, the variation in catechin profiles could also be due to differences in the growing climate, harvesting practices, postharvest storage, soil composition, and manufacturing practices [38]. 

C, CG, GC, and GCG are minor catechins in these tea samples, which account for only 0.10%–2.05%, 0.08%–2.11%, 0.18%–1.60%, and 0.15%–4.38%, respectively, of the total catechin content in green tea, yellow tea, white tea, and oolong tea. They were absent in black tea. The levels of C and EC were the highest in oolong tea samples.

### 3.3. Antioxidant Activity of Different Kinds of Tea Samples

Among the six samples investigated, green tea dust exhibited the highest antioxidant capacity to scavenge free radicals as well as to reduce Fe^3+^ (Figure 1B,C). In contrast, the antioxidant capacity of black tea was the lowest among all the samples. The values of total phenolic contents of the tea samples decreased in the order of GTD > YT > WT > GT > OT > BT (Figure 1A), while those of antioxidant capacity were in the order of GTD ≈ YT > GT ≈ WT > OT > BT (Figure 1B,C). Thus, the antioxidant capacities of the six tea samples seemed almost in proportion to the total phenolic content of the samples. Supportively, Pearson’s correlation coefficient analysis showed strong positive correlations between total phenolic contents and DPPH values (r = 0.919, *p* < 0.01) and between total phenolic contents and FRAP values (r = 0.945, *p* <0.01). This was consistent with the findings of Vinci et al. [39] and Anesini et al. [40]. Henning et al. [41] also reported a significant correlation between flavanol content in tea and oxygen radical absorbance capacity (ORAC) values.

### 3.4. Inhibition of Harmful Substances in Potato Chips by Pre-soaking Treatment with Different Tea Extracts

#### 3.4.1. Inhibition of HMF Formation

In potato chips prepared in our laboratory, the formation of HMF was initially 7.68 mg/kg. Pre-soaking treatment with the six tea extracts at a level of 30 g/L showed a significant inhibitory effect on its generation, varying from 19.8% to 53.2% (Figure 2A). Among the tea extracts tested, green tea dust and white tea exhibited the best inhibitory effect, which reduced HMF content in potato chips by 53.2% and 47.3%, respectively, whereas old green tea and oolong tea showed the worst inhibitory effects, with 23.5% and 19.9% reduction in HMF, respectively. Regarding the values of total phenolic contents and antioxidant capacities of the tea samples, no clear association was observed between these indices and the inhibitory effects of the tea extracts on HMF formation. The contribution of individual components to HMF inhibition might provide a more reasonable explanation. Catechins, especially EGCG, ECG, and EC, showed distinct mitigation effects on HMF in both chemical and food models [42]. They are adducted with HMF or its precursor, 3-deoxyglucosone, to reduce the formation of 5-HMF during food preparation [43,44]. The high contents of these catechins might contribute to the inhibitory effects of green tea extracts on HMF generation. Other than catechins, amino acids, such as cysteine, lysine, glycine, phenylalanine, and histidine, could also mitigate HMF by the formation of HMF-amino acid adducts [2,45,46,47]. Green tea, oolong tea, and yellow tea contain higher levels of cysteine than black and white tea, whereas the levels of lysine, glycine, phenylalanine, and histidine were relatively high in white tea compared to the other types of tea [16]. This might partially explain the distinguishing inhibitory effect of white tea extract on HMF formation in potato chips. The inhibitory effect of various tea extracts on harmful substances depends on the combined influence of the complex constituents in the extracts.

#### 3.4.2. Inhibition of GO and MGO Formation

As shown in Figure 2B,C, the tea extracts significantly reduced the contents of GO and MGO in potato chips. The six tea extracts showed a similar extent of inhibition of GO, ranging from 26.9% to 36.6%, while in the case of MGO, the inhibitory effects varied remarkedly between different tea samples. The two green tea extracts and the black tea extract showed the best inhibitory capacity for MGO, with 68.2%–75.1% reductions. Yellow tea and white tea, in contrast, displayed the worst inhibitory effects, only reducing MGO by 16.1% and 19.5%, respectively. The performance of tea extracts for MGO inhibition rate followed the order of BT > GTD > GT > OT > WT > YT, which was different from their performance for HMF inhibition. Different inhibitory mechanisms might contribute to the distinct mitigation effects of the tea extracts on each individual harmful compound. Green tea extracts contain high levels of catechins, which possess efficient MGO scavenging capacity [48]. However, black tea extract contained a very small amount of these catechins, while it exhibited the highest efficiency for the inhibition of MGO in this study. In black tea, theaflavins are the major phenolic compounds that are derived from the enzymatic oxidation of catechins during fermentation. Theaflavins were revealed to exert a prominent elimination capacity on MGO by interaction and formation of MGO-substituted theaflavins [49]. Moreover, the higher levels of free amino acids possessing efficient scavenging capacities for MGO in black tea might contribute further to the elimination of MGO [3,50]. Together, these might explain the remarkable inhibitory effect of black tea extract on the level of MGO in potato chips [49].

### 3.5. Inhibitory Effects of Different Concentrations of Tea Extracts on the Generation of Diverse Harmful Substances

GO, MGO, and HMF are important endogenous hazardous substances inevitably generated during the thermal processing of foods. Moreover, they are also prominent precursors for the formation of acrylamide and AGEs in food [3,51]. Considering the great inhibitory potential of green tea dust, old green tea (stored over a year), and black tea on the formation of GO, MGO, and HMF, they were selected for subsequent investigations of the simultaneous inhibitory effects of different concentrations of tea extracts on the formation of diverse harmful compounds, namely, GO, MGO, HMF, acrylamide, and AGEs, in potato chips. The carbonyl values in the oil of potato slices, which can also reflect the quality of potato chips, were evaluated. Considering the preference of customers, the influences of tea extracts on the color and texture of chips were also evaluated.

#### 3.5.1. Reduction in Carbonyl Value in the Oil Extract of Potato Chips

Carbonyl value (CV) is an indicator of harmful compounds such as ketones and aldehydes generated by the oxidation of fats and oils at high processing temperatures and indicates the degree of food rancidity and reduced nutritional value [52]. Thus, we extracted the oils absorbed in the potato chips and analyzed the CV of these extracts. As shown in Figure 3, all the tea extracts significantly improved the quality of fried potato chips (*p* < 0.05). The two green tea extracts showed better effects than the black tea extract, with a reduction in the CV of 35.7%–38.7% after the pre-soaking treatment. Thus, the three tea extracts all showed potential for the control of hazardous compounds in thermally processed potato chips. Then, we investigated further in-depth their inhibitory effects on different individual harmful compounds that were simultaneously generated during the production of potato chips.

#### 3.5.2. Inhibition of HMF

As shown in Figure 4A, the generation of HMF in the potato chips was inhibited considerably in a dose-dependent manner by tea extracts. The higher the concentration of the tea extracts, the greater the inhibition rate of HMF formation. Again, the green tea dust exhibited the best inhibitory capacity for HMF formation among the three tea samples tested. The chips pre-soaked with 50 g/L green tea dust extract reduced HMF by 73.3%, whose concentration reached the lowest at 2.03 mg/kg of all the samples tested. Although the total phenolic content and the antioxidant activity of black tea were the lowest in all the tea samples selected for this study, they showed a remarkable inhibitory capacity for HMF in potato chips. The application of 50 g/L black tea extract decreased HMF in the final potato chip products by 43.8%. As discussed above, other components, such as amino acids and theaflavins, might be involved in the reduction mechanism of HMF by the pre-soaking treatment of black tea extract.

#### 3.5.3. Inhibition of GO and MGO

In contrast to the findings of HMF, no clear dose-dependent trends were observed on the inhibitory effects of the three tea extracts on the formation of GO and MGO (Figure 4B,C). The highest inhibitory effect on GO formation was observed at the level of 30 g/L in all the tea samples. A further increase in the concentration resulted in a significant reduction in the inhibitory capacity. The inhibition of MGO in potato chips also reached a maximum at the pre-soaking level of 30 g/L (Figure 4C). When the concentration of the tea extracts further increased from 30 g/L to 50 g/L, the MGO content started to increase by 20.0%, 36.2%, and 7.3% in potato chips treated with green tea dust, old green tea, and black tea, respectively. GO and MGO, representing the typical dicarbonyl compounds, are the highly reactive intermediates produced in the reactions of Maillard reaction, caramelization, and lipid peroxidation. They also act as major precursors of various harmful components, such as acrylamide, heterocyclic amine, and AGEs [3,53]. Thus, both the formation and consumption of dicarbonyl compounds occurred simultaneously during food processing, which should be considered and investigated together in-depth in the future.

#### 3.5.4. Inhibition of Acrylamide

Except for the chips treated with 10 g/L black tea extract, the generation of acrylamide was markedly inhibited by pre-soaking with tea extracts (Figure 4D). One-year-stored green tea extract displayed the best inhibitory effect on acrylamide formation in a dose-dependent manner. It inhibited the generation of acrylamide in potato chips by 24.1% at the treatment level of 50 g/L, while 50 g/L green tea dust extract also decreased the acrylamide level in potato chips by 21.8%. Thus, both the green tea dust and the old green tea were effective candidates for preventing the formation of acrylamide in thermally processed potato chips. This might be attributed to the high concentration of catechins in green tea samples. Catechins were very efficient trapping agents of reactive carbonyl species, which in turn reduces the formation of acrylamide. They may also inhibit acrylamide formation through their antioxidant activities to prevent lipid oxidation [54]. Again, the inhibitory effect of black tea extract on acrylamide formation might be mostly attributed to the trapping of dicarbonyl compounds, its reactive precursors, by theaflavins and amino acids abundant in black tea [3,49,50].

#### 3.5.5. Inhibition of AGEs and Protein Oxidative Products

Catechins were capable of lowering the concentration of AGEs in foods and in vivo via various mechanisms, such as the reduction in HMF formation, the elimination of reactive carbonyl species through adduct formation, and the elimination of free radicals [48]. As a result, the extracts of two green teas displayed an efficient inhibitory effect on the levels of AGEs and protein oxidative products in fried potato chips in a clear dose-dependent manner. (Figure 5). The reduction in AGEs, dityrosine, kynurenine, and N’-formylkynurenine reached the maximum of 63.8%, 63.7%, 66.1%, and 64.2%, respectively, when pre-soaked with 50 g/L extract of one-year-stored green tea, and reached the maximum by 42.9%, 41.6%, 55.0%, and 43.2% when pretreated with 50 g/L green tea dust extract. In contrast, black tea extract exhibited the best inhibitory capacity at the treatment level of 10 g/L, whereas the weakest effect was observed at the level of 50 g/L. This might be caused by the increased migration of self-contained AGEs in the black tea extract to the potato chips when immersing the chips in a more concentrated tea extract. Jiao et al. [55] found that black tea showed higher levels of AGEs (CML and CEL) than green and oolong teas. The main pathways involved in the formation of AGEs during black tea processing were more likely to be the fructoselysine pathway but not the GO/MGO pathway [56].

### 3.6. Impact of Pre-Soaking Treatment with Tea Extract on Color and Texture of Potato Chips

Color and texture are important attributes that determine consumers’ acceptance of and preference for products. Table 2 shows the changes in L*, a*, and b* values of the potato chips after treatment with different concentrations of tea extracts. As the concentration of tea extract increased, the L* value of potato chips decreased, which indicated that the potato chips turned darker. Among the three kinds of tea tested, the potato chips pre-soaked in black tea extract were the darkest, and the chips pretreated with the one-year-stored green tea extract were the brightest. The a* value indicates the degrees of green (−)/red (+) color. It increased as the potato chips were pretreated with black tea extract but decreased when they were pretreated with the two green tea extracts. The b* value indicates the degrees of blue (−)/yellow (+) color. The potato chips pretreated with different tea extracts all showed decreases in b* values in comparison with the blank sample, indicating that the products turned blue. Given consumers’ preference for a typical yellow-bright color of potato chip products, green tea extracts should be a better choice than black tea extract.

In relation to texture, the value of fracturability of potato chips all increased to the highest when treated with 10 g/L extracts of different teas but decreased further to the statistically equal level of the control group as the tea extract concentration increased to 50 g/L. Therefore, a higher concentration (30–50 g/L) of tea extract would better retain the texture of the fried potato chips. Further investigations on the effects of tea extract treatments on the moisture retention, oil absorption, coagulation of the proteins, and gelatinization of the starch [57,58] might provide a satisfactory explanation of the phenomena observed in our study.

## 4. Conclusions

Among the six tea extracts investigated, the extract of green tea dust waste exhibited the most promising inhibitory effects on the typical harmful substances in fried potato chips. The contents of HMF, GO, MGO, acrylamide, dityrosine, kynurenine, *N’*-formylkynurenine, and AGEs in fried potato chips were reduced by 73.3%, 20.3%, 69.7%, 21.8%, 41.6%, 55.0%, 43.2%, and 42.9%, respectively, after pre-soaking of the fresh potato slices with green tea dust extract at 50 g/L. Meanwhile, the color and texture were affected by the treatment the least. This suggested that the utilization of green tea dust for the control of harmful substances in fried potato chips is practical and valuable. Total phenolic content and antioxidant capacity showed great differences between the tea extracts, whereas no clear association was observed between them and the inhibitory effects on the harmful substances. Previous investigations indicated that catechins, theaflavins, and amino acids all showed an inhibitory capacity for diverse thermal-processing-derived hazardous compounds. Therefore, a combination of the inhibitory effects of these ingredients, which are distributed unevenly in different teas, were supposed and deserve further investigation. The utilization of green tea dust, a predominant waste from the tea industry, especially in Asian countries, for the improvement of food quality and safety in other food production is also of sound and practical importance.

## Figures and Tables

**Figure 1 foods-12-00321-f001:**
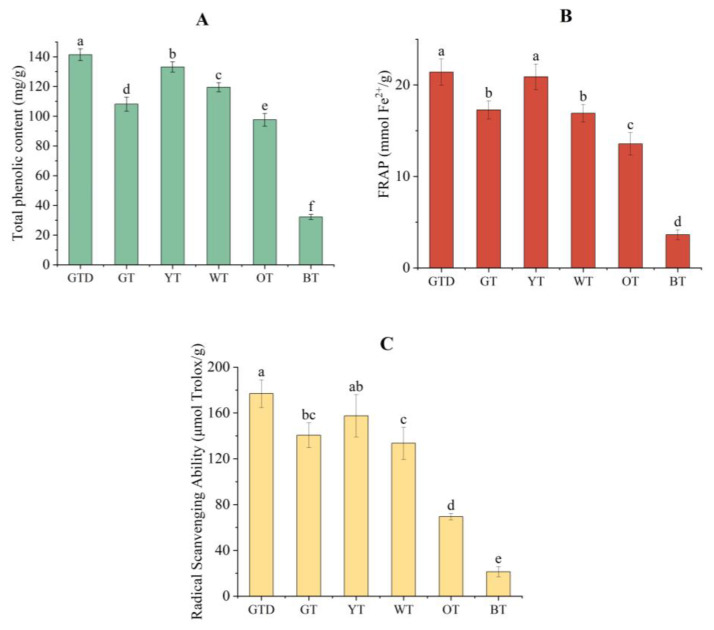
The total phenolic content (**A**) and antioxidant activities (**B**, FRAP assay; **C**, DPPH assay) of different tea samples. Different letters in a plot indicate significant differences between samples (*p* < 0.05). Abbreviations: GTD, green tea dust; GT, green tea stored over one year; YT, yellow tea; WT, white tea; OT, oolong tea; BT, black tea.

**Figure 2 foods-12-00321-f002:**
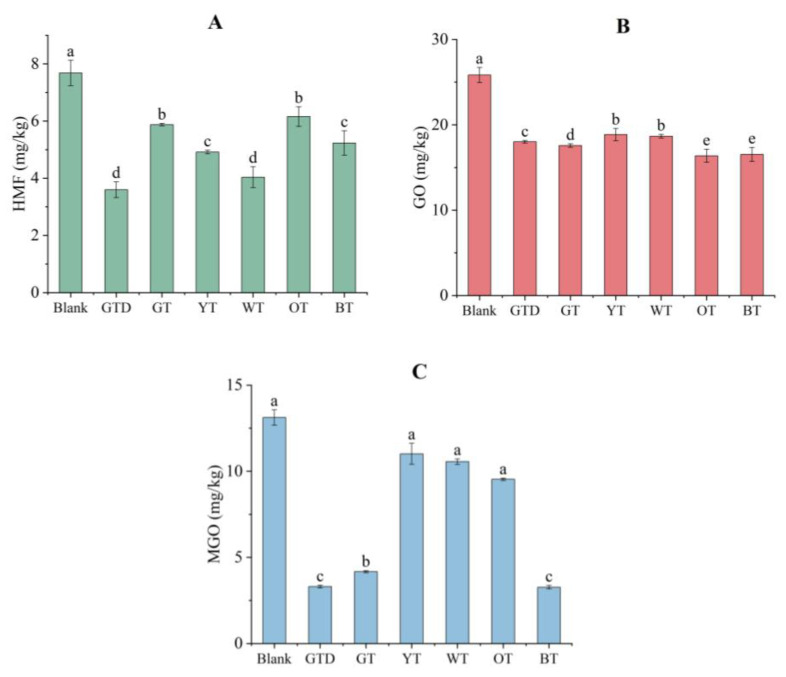
Contents of HMF (**A**), GO (**B**), and MGO (**C**) in potato chips pre-soaked in different tea extracts (30 g/L). The blank sample was pre-soaked in distilled water prior to frying. Different letters in a plot indicate significant differences between samples (*p* < 0.05). Abbreviations: GTD, green tea dust; GT, green tea stored over one year; YT, yellow tea; WT, white tea; OT, oolong tea; BT, black tea.

**Figure 3 foods-12-00321-f003:**
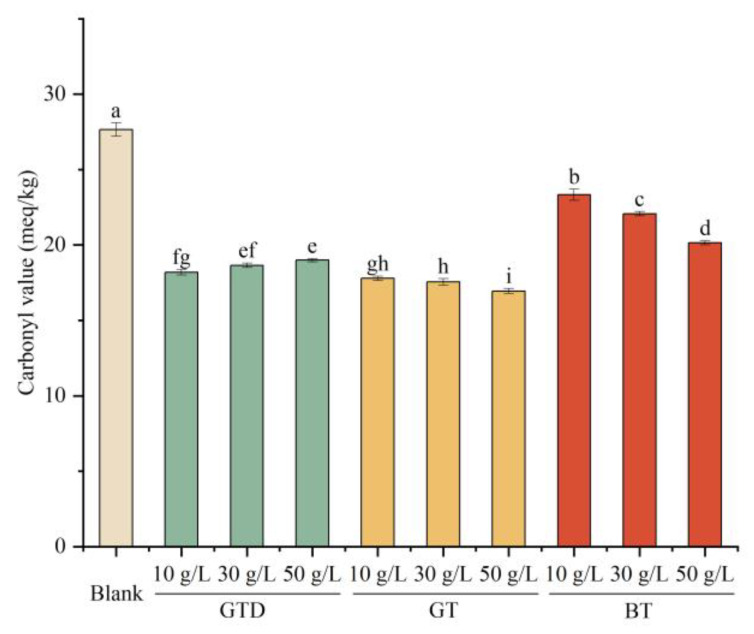
Comparison of carbonyl values of potato chips pre-soaked in different tea extracts. Different letters indicate significant differences between samples (*p* < 0.05). Abbreviations: GTD, green tea dust; GT, green tea stored over one year; BT, black tea.

**Figure 4 foods-12-00321-f004:**
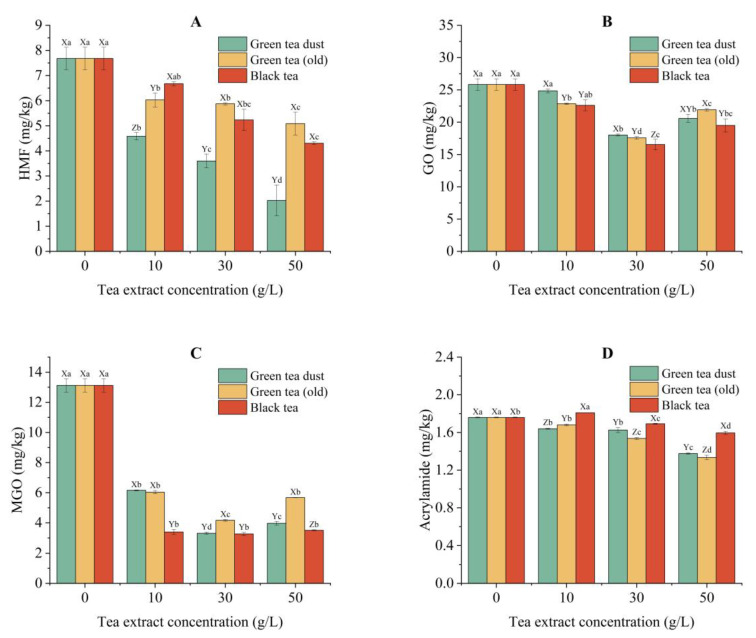
Inhibitory effects of different concentrations (10 g/L, 30 g/L, and 50 g/L) of tea extracts on the generation of HMF (**A**), GO (**B**), MGO (**C**), and acrylamide (**D**) in potato chips. Different capital letters (X–Z) indicate significant differences (*p* < 0.05) between different tea extracts of the same concentration. Different lowercase letters (a-d) indicate significant differences (*p* < 0.05) between various concentrations of the same tea extract.

**Figure 5 foods-12-00321-f005:**
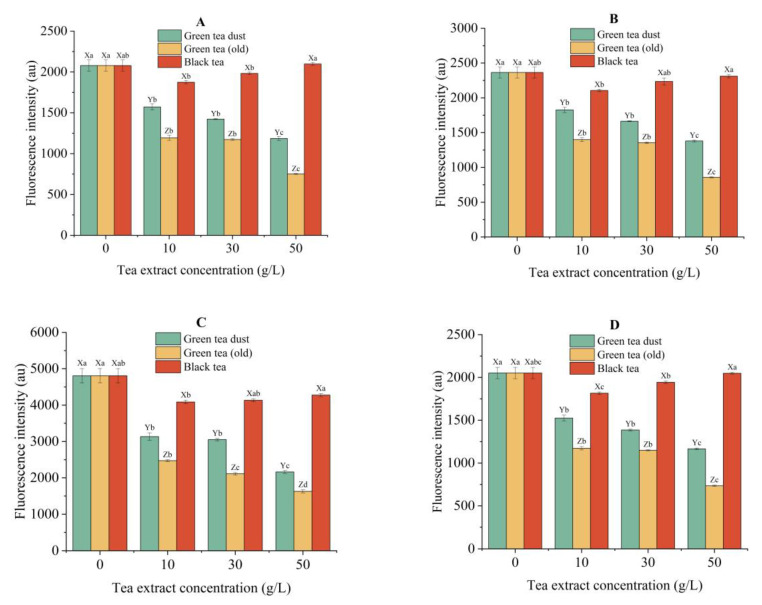
Inhibitory effects of different concentrations (10 g/L, 30 g/L, and 50 g/L) of tea extracts on the levels of AGEs (**A**), dityrosine (**B**), kynurenine (**C**), and *N*’-formylkynurenine (**D**) in potato chips. Different capital letters (X–Z) indicate significant differences (*p* < 0.05) between different tea extracts of the same concentration. Different lowercase letters (a–d) indicate significant differences (*p* < 0.05) between various concentrations of the same tea extract.

**Table 1 foods-12-00321-t001:** Contents (mg/g) of catechins in different tea samples.

Compound	Green Tea Dust	Green Tea (Old)	Yellow Tea	White Tea	Oolong Tea	Black Tea
C	0.68 ± 0.11 bcK	1.18 ± 0.05 cJ	2.41 ± 0.07 dI	0.79 ± 0.01 bcK	2.63 ± 0.09 eH	ND
EC	2.50 ± 0.26 bcIJ	9.38 ± 0.49 bH	11.88 ± 0.24 cH	4.05 ± 0.24 bI	13.00 ± 0.20 cH	0.11 ± 0.02 abJ
CG	0.12 ± 0.01 cH	0.10 ± 0.01 dH	0.10 ± 0.01 fH	0.09 ± 0.01 dH	0.09 ± 0.01 gH	ND
ECG	21.85 ± 2.79 abcHIJKL	9.91 ± 0.26 bJ	17.39 ± 0.27 bH	14.79 ± 0.42 aI	4.04 ± 0.12 dK	0.10 ± 0.01 bL
EGC	10.50 ± 0.91 bcK	39.95 ± 1.17 aK	44.03 ± 0.62 aH	7.29 ± 0.64 bcdL	21.79 ± 0.18 aJ	0.16 ± 0.02 abM
GC	1.03 ± 0.12 bcJ	2.00 ± 0.22 cdI	2.64 ± 0.29 defH	0.64 ± 0.10 cdJ	2.28 ± 0.22 defgHI	ND
EGCG	48.58 ± 0.20 aH	33.95 ± 0.52 aJ	45.89 ± 0.23 aI	22.28 ± 1.23 aK	15.42 ± 0.32 bK	0.18 ± 0.01 aL
GCG	0.83 ± 0.01 bHI	0.86 ± 0.02 cHI	0.83 ± 0.02 eHI	0.81 ± 0.01 bcH	0.79 ± 0.02 fI	ND
Total catechin content	86.09 ± 3.57 J	97.33 ± 2.55 I	125.18 ± 1.24 H	50.75 ± 2.19 L	60.05 ± 0.90 K	0.58 ± 0.05 M

Different lowercase letters (a–g) within a column indicate significant differences (*p* < 0.05) between different catechins in a tea sample. Different capital letters (H–M) within a row indicate significant differences (*p* < 0.05) between different tea samples. ND represents not detected.

**Table 2 foods-12-00321-t002:** The color parameters (L*, a*, and b* values) and fracturability of potato chips pre-soaked in three tea extracts at different concentrations.

Parameters		Sample	0 g/L	10 g/L	30 g/L	50 g/L
Color	L*	Green tea dust	60.30 ± 2.20 a	57.46 ± 0.73 aY	50.68 ± 1.50 bY	50.07 ± 2.33 bY
		Green tea (old)	60.30 ± 2.20 a	60.24 ± 0.97 aX	62.67 ± 1.27 aX	55.50 ± 1.39 bX
		Black tea	60.30 ± 2.20 a	49.85 ± 1.46 bZ	47.33 ± 1.35 bcZ	44.60 ± 1.00 cZ
	a*	Green tea dust	13.40 ± 0.80 a	11.00 ± 0.67 aY	12.06 ± 0.15 aXY	12.33 ± 0.05 aXY
		Green tea (old)	13.40 ± 0.80 a	8.84 ± 0.45 bZ	9.92 ± 0.20 abY	12.30 ± 0.29 aY
		Black tea	13.40 ± 0.80 b	14.00 ± 0.67 abX	15.27 ± 1.03 aX	15.15 ± 0.48 aX
	b*	Green tea dust	36.00 ± 2.60 a	33.87 ± 1.11 abX	30.60 ± 1.92 bXY	31.20 ± 1.07 bX
		Green tea (old)	36.00 ± 2.60 a	33.07 ± 0.96 abX	32.13 ± 1.11 bX	31.30 ± 1.27 bX
		Black tea	36.00 ± 2.60 a	33.93 ± 0.78 aX	27.15 ± 1.13 bY	28.30 ± 2.21 bX
Fracturability (g)		Green tea dust	75.00 ± 9.90 b	122.00 ± 6.38 aY	85.33 ± 7.61 bX	76.67 ± 2.49 bX
		Green tea (old)	75.00 ± 9.90 b	106.00 ± 4.32 aZ	85.25 ± 4.66 bX	80.33 ± 7.70 bX
		Black tea	75.00 ± 9.90 b	211.00 ± 7.87 aX	78.50 ± 8.65 bX	83.00 ± 2.68 bX

Different lowercase letters (a–c) indicate significant differences (*p* < 0.05) between different extract concentrations. Different capital letters (X–Z) indicate significant differences (*p* < 0.05) between different tea samples.

## Data Availability

Data is contained within the article or Appendix A.

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
