# Peer review of "Reduction in Five Harmful Substances in Fried Potato Chips by Pre-Soaking Treatment with Different Tea Extracts"

_foods, 2023, doi:10.3390/foods12020321_

Round 1

Reviewer 1 Report

The paper concerns the use of tea extracts as a soaking solution to prevent acrylamide and other potentially harmful compounds in deep-fried potato chips. The topic is interesting for food scientists, many thermal damage parameters were evaluated and various tea extracts in different concentrations were tested. The work is scientifically sound but the paper has important defects that must be corrected in order to become suitable for publication. Here below the main points that need to be addressed.

- The English language must be fully revised, it is sometimes difficult to understand the meaning of sentences and misunderstandings can easily occurr. 

- Abstract: mention all types of tea tested in the work. You only give results about green tea dust, some results (maybe ranges or min-max) about the other extracts should be given.

- Keywords: delete “reduction” from the keywords.

- Introduction: some information about tea types and their phenolic and antioxidant profile should be added in the introduction. What is said in lines 200-204 and 212-222, concerning tea production process and differences between the different types of tea should be moved in the introduction.  There is total lack of reference to literature papers about the use of tea extracts for the prevention of acrylamide or other thermal damage indices in heat-treated foods (see for instance: Interaction of dough preparation method, green tea extract and baking temperature on the quality of rye bread and acrylamide content. Onacik-Gur, S., Szafranska, A., Roszko, M., Stepniewska, S. LWT -- Food Science and Technology. 154, 112759, 2022; Quality characteristics, antioxidant activity, and acrylamide content of cookies made with powdered green tea. Eun-Sun Hwang, Tae Young Park. Journal of the Korean Society of Food Science and Nutrition. 50, (10): 1082-1090; Inhibitory effect of natural antioxidants on the formation of thermal processing hazards in red braised pork. Liu Lichun, Jiang Yujie, Shen Mingyue, Xie Mingyong, Nie Shaoping. Food Science, China. 42, (15): 50-57, 2021; Effect of the addition of soluble dietary fiber and green tea polyphenols on acrylamide formation and in vitro starch digestibility in baked starchy matrices. Torres, J. D., Dueik, V., Carre, D., Bouchon, P. Molecules. 24, (20): 3674, 2019; Effect of (-)-epigallocatechin gallate (EGCG) extracted from green tea in reducing the formation of acrylamide during the bread baking process. Zhengjie Fu, Yoo, M. J. Y., Weibiao Zhou, Lei Zhang, Yutao Chen, Jun Lu. Food Chemistry. 242, 162-168, 2018; Microwave thawing and green tea extract efficiency for the formation of acrylamide throughout the production process of chicken burgers and chicken nuggets. Soncu, E. D., Kolsarici, N. Journal of the Science of Food and Agriculture. 97, (6): 1790-1797, 2017; Effect of green tea extract and microwave pre-cooking on the formation of acrylamide in fried chicken drumsticks and chicken wings. Demirok, E., Kolsarici, N. Food Research International. 63, (Part C, 2nd Conference on Coffee Cocoa and Tea Science (CoCoTea2013)): 290-298, 2014; Effect of natural extracts on the formation of acrylamide in fried potatoes. Morales, G., Jimenez, M., Garcia, O., Mendoza, M. R., Beristain, C. I. LWT -- Food Science and Technology. 58, (2): 587-593, 2014; Influence of addition of green tea and green coffee extracts on the properties of fine yeast pastry fried products. Budryn, G., Zyzelewicz, D., Nebesny, E., Oracz, J., Krysiak, W. Food Research International. 50, (1): 149-160, 2013.) The state-of-art must be considered in the introduction and/or in the discussion.

- Materials and Methods: the experimental procedures and methods are poorly described.

For instance, line 93-94: all the time at 100°C? How was the solution filtered (type of filter)? Line 99: 7,5% (v/v)? Natrium carbonate is solid.  

For all the analytical determinations, how many replicates were carried out? Data are expressed as mean value and the number of replicates must be indicated.

Line 105: add that the determination was conducted on the 10 g/L tea extract (I guess)

Line 126 onwards: indicate the temperature of the soaking solutions.

Line 135 onwards: color was measured on five slices or 5 times on the same slice?

Line 143: what parameter of the peak is considered as fracturability (area, maximum force, height)?

Line 144 onwards: rewrite the method, three subsequent extractions were carried out (not triplicated), volume of water?

Line 156: 5% acqueous methanol is not clear, better say water:methanol 95:5 (v/v) , the same in line 165.

Line 186 onwards: the method must be better described, the cited reference is not explicative.

Line 192 onwards: as mentioned above, the number of replicates used to calculate the mean values and to process data by ANOVA must be indicated.

Results and discussion: as already mentioned, information about tea processing must be moved to the introduction.

Legend to Figure 1 (line 230): add that the data refer to tea extract 10 g/L.

Line 243-245: add a reference.

Paragraph 3.2 (line 234 onwards): there isn’t any comparison with data from the literature about the content of catechins in the various teas; the extremely low amount of catechins in the black tea extract deserves some comments.

Title of Table 1 (line 247): contents (mg/g) of catechins in different tea extracts (10 g/L)

Paragraph 3.3 (line 271 onwards): refer to figure 1 (B and C).

Lines 286-289: there is no statistically significant difference between GT and WT, and there is a clear correlation between total phenolic content and antioxidant activity, therefore the statement should be changed and the correlation could be studied. Here as well literature data can help.

Line 320-321: rewrite, difficult to understand.

Line 330:  the word “effort” makes no sense

Line 341: add the carbonyl value in the oil

Paragraph 3.5.1. Explain that this parameter is referred to the oil absorbed in the potato chips.

Line 356: remove yellow, white and oolong tea from the caption, they are not considered.

Line 363: remove p<0.05, inappropriate.

Figure 4: it would be better to show raw data instead of % reduction: it makes no senso to apply statistics at zero value, and the statistical analysis would be clearer, without negative values. Add the letter “d” in line 375 since it is present in the graphs.

Line 384-385: 1.8%-11.5% reduction with respect to…?

Line 385-386: rewrite, the meaning is not clear.

Line 389-391: rewrite, the meaning is not clear.

Line 429: add “d” in the caption.

Line 436:      the statement is not justified, b* values are statistically similar at 30 and 50 g/L for all teas.

Line 445-448: some explanation or hypothesis about the effects of different extract concentrations on the texture should be given.

Line 457: I suggest to modify “the best” into “the most promising”.

Line 465-468: rewrite, difficult to understand.

As a conclusion, the paper needs substantial improvements.

Author Response

Dear Reviewer,

Thank you very much for the valuable comments on our manuscript (foods-2095775). Based on these comments and suggestions, we have made careful modifications of the manuscript. The manuscript was uploaded for your re-consideration. The corrections in the manuscript were marked in red, and the responses to the comments are listed below. We hope the manuscript is now suitable for publication in Foods.

Sincerely yours,

Jie Zheng

Response to the comments

The paper concerns the use of tea extracts as a soaking solution to prevent acrylamide and other potentially harmful compounds in deep-fried potato chips. The topic is interesting for food scientists, many thermal damage parameters were evaluated and various tea extracts in different concentrations were tested. The work is scientifically sound but the paper has important defects that must be corrected in order to become suitable for publication. Here below the main points that need to be addressed.

- The English language must be fully revised; it is sometimes difficult to understand the meaning of sentences and misunderstandings can easily occur.

Response: We sent the manuscript for the editing service by MDPI for language correction. We greatly appreciate your comment.

- Abstract: mention all types of tea tested in the work. You only give results about green tea dust, some results (maybe ranges or min-max) about the other extracts should be given.

Response: Thanks for your advice. We have modified our abstract accordingly.

- Keywords: delete “reduction” from the keywords.

Response: Thank you for your suggestion. It was deleted.

- Introduction: some information about tea types and their phenolic and antioxidant profile should be added in the introduction. What is said in lines 200-204 and 212-222, concerning tea production process and differences between the different types of tea should be moved in the introduction. 
There is total lack of reference to literature papers about the use of tea extracts for the prevention of acrylamide or other thermal damage indices in heat-treated foods (see for instance: (1) Interaction of dough preparation method, green tea extract and baking temperature on the quality of rye bread and acrylamide content. Onacik-Gur, S., Szafranska, A., Roszko, M., Stepniewska, S. LWT -- Food Science and Technology. 154, 112759, 2022; (2) Quality characteristics, antioxidant activity, and acrylamide content of cookies made with powdered green tea. Eun-Sun Hwang, Tae Young Park. Journal of the Korean Society of Food Science and Nutrition. 50, (10): 1082-1090; (3) Inhibitory effect of natural antioxidants on the formation of thermal processing hazards in red braised pork. Liu Lichun, Jiang Yujie, Shen Mingyue, Xie Mingyong, Nie Shaoping. Food Science, China. 42, (15): 50-57, 2021; (4) Effect of the addition of soluble dietary fiber and green tea polyphenols on acrylamide formation and in vitro starch digestibility in baked starchy matrices. Torres, J. D., Dueik, V., Carre, D., Bouchon, P. Molecules. 24, (20): 3674, 2019; (5) Effect of (-)-epigallocatechin gallate (EGCG) extracted from green tea in reducing the formation of acrylamide during the bread baking process. Zhengjie Fu, Yoo, M. J. Y., Weibiao Zhou, Lei Zhang, Yutao Chen, Jun Lu. Food Chemistry. 242, 162-168, 2018; (6) Microwave thawing and green tea extract efficiency for the formation of acrylamide throughout the production process of chicken burgers and chicken nuggets. Soncu, E. D., Kolsarici, N. Journal of the Science of Food and Agriculture. 97, (6): 1790-1797, 2017; (7) Effect of green tea extract and microwave pre-cooking on the formation of acrylamide in fried chicken drumsticks and chicken wings. Demirok, E., Kolsarici, N. Food Research International. 63, (Part C, 2nd Conference on Coffee Cocoa and Tea Science (CoCoTea2013)): 290-298, 2014; (8) Effect of natural extracts on the formation of acrylamide in fried potatoes. Morales, G., Jimenez, M., Garcia, O., Mendoza, M. R., Beristain, C. I. LWT -- Food Science and Technology. 58, (2): 587-593, 2014;  (9) Influence of addition of green tea and green coffee extracts on the properties of fine yeast pastry fried products. Budryn, G., Zyzelewicz, D., Nebesny, E., Oracz, J., Krysiak, W. Food Research International. 50, (1): 149-160, 2013.)
The state-of-art must be considered in the introduction and/or in the discussion.

Response: We greatly appreciate your comments. The description of tea types and their phenolic and antioxidant profile (Line 200-204 in original version) were moved into introduction part. However, we feel the sentences in Line 212-222 (in original version) should be better to stay in the discussion part as we used them to discuss the differences in phenolic profiles of the six tea samples investigated in this study. Finally, thank you for providing us the useful references. We have added some of them in the introduction part and discussed more about the current researches on the use of tea extracts for the prevention of acrylamide and other harmful substances in foods.

- Materials and Methods: the experimental procedures and methods are poorly described.

Response: Thanks for your suggestion. With your guidance, we have carefully rewritten this part.

For instance, line 93-94: all the time at 100°C? How was the solution filtered (type of filter)? Line 99: 7,5% (v/v)? Natrium carbonate is solid. 

Response: Thanks a lot. We have corrected these and added the details for understanding.

For all the analytical determinations, how many replicates were carried out? Data are expressed as mean value and the number of replicates must be indicated.

Response: We greatly appreciate for your comment. We have added to the number of repetitions of the experiment in the part of Statistical Analysis.

Line 105: add that the determination was conducted on the 10 g/L tea extract (I guess)

Response: Thanks for your comment. We have measured the antioxidant properties of different tea extracts at 10 g/L and have added the information in line 145.

Line 126 onwards: indicate the temperature of the soaking solutions.

Response: Thank you so much. We have indicated the temperature of the soaking solutions in line 169.

Line 135 onwards: color was measured on five slices or 5 times on the same slice?

Response: Thanks for your question. We measured the color on five slices. And the relative information has been added into Color and Texture Analysis part and Statistical Analysis part.

Line 143: what parameter of the peak is considered as fracturability (area, maximum force, height)?

Response: Thanks for your question. The force corresponding to that peak is defined as fracturability. We have added the information from line 186 to 187 to make the expression clearer.

Line 144 onwards: rewrite the method, three subsequent extractions were carried out (not triplicated), volume of water?

Response: We greatly appreciate for your comment. We have re-described the method in the section 2.8.

Line 156: 5% acqueous methanol is not clear, better say water:methanol 95:5 (v/v) , the same in line 165.

Response: Thanks for your comment. We have changed it in line 203 and 211.

Line 186 onwards: the method must be better described, the cited reference is not explicative.

Response: Thank you so much. We have described the method in more detail in line 234-243.

Line 192 onwards: as mentioned above, the number of replicates used to calculate the mean values and to process data by ANOVA must be indicated.

Response: Thanks a lot. We have added to the number of repetitions of the experiment from line 245 to 247.

Results and discussion: as already mentioned, information about tea processing must be moved to the introduction.

Response: Thank you. We have moved these to Introduction part.

Legend to Figure 1 (line 230): add that the data refer to tea extract 10 g/L.

Response: Thanks a lot. Actually, we calculated the data on basis of the weight of tea samples (g). We add the explanation in the Method part (line 145-146)

Line 243-245: add a reference.

Response: Thanks for your comment. We have added the reference in line 311.

Paragraph 3.2 (line 234 onwards): there isn’t any comparison with data from the literature about the content of catechins in the various teas; the extremely low amount of catechins in the black tea extract deserves some comments.

Response: Thank you for your suggestion. Some related literatures were added and discussed in line 296-306.

Title of Table 1 (line 247): contents (mg/g) of catechins in different tea extracts (10 g/L)

Response: Thanks a lot. We have corrected the expression to “Contents (mg/g) of catechins in different tea samples” in line 312.

Paragraph 3.3 (line 271 onwards): refer to figure 1 (B and C).

Response: Thanks a lot. We referred to these two figures in line 339.

Lines 286-289: there is no statistically significant difference between GT and WT, and there is a clear correlation between total phenolic content and antioxidant activity, therefore the statement should be changed and the correlation could be studied. Here as well literature data can help.

Response: Thank you for the comment. The information given in previous line 286-288 were moved to Section 3.3 for better discussion of the relationship between total phenolic content and antioxidant activity, and more references were cited for support.

Line 320-321: rewrite, difficult to understand.

Response: Thank you for your suggestion. We changed this no-sense sentence from line 359 to 361.

Line 330:  the word “effort” makes no sense

Response: Thank you for your suggestion. It was corrected in line 400.

Line 341: add the carbonyl value in the oil

Response: Thanks a lot. We have added the description from line 412 to 414.

Paragraph 3.5.1. Explain that this parameter is referred to the oil absorbed in the potato chips.

Response: Thank you for your suggestion. Explanation has been added in line 415.

Line 356: remove yellow, white and oolong tea from the caption, they are not considered.

Response: Thank you very much. We have corrected this error.

Line 363: remove p<0.05, inappropriate.

Response: Thank you so much. It was removed.

Figure 4: it would be better to show raw data instead of % reduction: it makes no senso to apply statistics at zero value, and the statistical analysis would be clearer, without negative values. Add the letter “d” in line 375 since it is present in the graphs.

Response: Thanks for your great advice. We have changed display format of Figure 4 after line 443. The statistical analysis has also been re-conducted.

Line 384-385: 1.8%-11.5% reduction with respect to…?

Response: We rewrote the 3.5.3 section. Thank you.

Line 385-386: rewrite, the meaning is not clear.

Response: This sentence was rewritten. Thank you.

Line 389-391: rewrite, the meaning is not clear.

Response: Thanks for your suggestion. This part was rewritten.

Line 429: add “d” in the caption.

Response: Thanks for your comment. It was added in line 504.

Line 436: the statement is not justified, b* values are statistically similar at 30 and 50 g/L for all teas.

Response: Thank you. The statements were corrected in line 510-512.

Line 445-448: some explanation or hypothesis about the effects of different extract concentrations on the texture should be given.

Response: Thank you for the suggestion. We have discussed and pointed the hypothesis on future studies from line 526 to 529.

Line 457: I suggest to modify “the best” into “the most promising”.

Response: We changed the expression in line 538. Thank you.

Line 465-468: rewrite, difficult to understand.

Response: Thank you. It was rewritten after line 546.

As a conclusion, the paper needs substantial improvements.

Reviewer 2 Report

Dear Authors,

Detailed notes on the manuscript are given below:

1) In "Abstract" (first sentence) and in "Introduction" (final part) the purpose of the work should be included.

2) Introduction: Alternative methods of modifying the acrylamide content in fried products are also known: DOI10.3390/su12083426

3) Line 71 - what kind of potato? (intended for fried products, potato chips?)

4) Line 93 – unit; liter [l], applies to the entire manuscript (ml, g/l, etc.) - SI system

5) Line 125 - Table A1; format inconsistent with MDPI editorial guidelines, applies to the entire manuscript

6) 2.14. Statistical Analysis - you used Duncan's post-hoc test, this means that the ANOVA was parametric. Provide the result of testing the normality of the distribution of the data population, the result of testing the homogeneity of variance in the samples (these are the conditions necessary for the use of parametric tests), specify what procedures you used and what was their result.

7) 2.14. Statistical Analysis - add information that Duncan's post-hoc test was used to determine homogeneous groups

8) Figure 1 - markings a-e; indicate that these are groups of homogeneous variables (refers to other figures and tables)

9) Clarify the methodology for measuring the color of the fried product DOI10.3390/su12083487

10) Figure 4 and 5. Inhibitory effects … - in my opinion, points on the graph should not be connected with a solid line (this suggests data prediction). If you want to visualize the course of the process, use a trend line (regression curve, least squares method)

11) 4. Conclusions ... I suggest changing it to "Summary"

The manuscript requires supplementation in methodological issues and the presentation of results

Author Response

Dear Reviewer,

Thank you very much for the valuable comments on our manuscript (foods-2095775). Based on these comments and suggestions, we have made careful modifications of the manuscript. The manuscript was uploaded for your re-consideration. The corrections in the manuscript were marked in red, and the responses to the comments are listed in the attached Word file. We hope the manuscript is now suitable for publication in Foods.

Sincerely yours,

Jie Zheng

Round 2

Reviewer 2 Report

Dear Authors, thank for correction of manuscript. 

Author Response

Dear Reviewer,

Thank you very much for the valuable comments on our manuscript (foods-2095775). We have made careful modifications of the manuscript. The manuscript was uploaded for your re-consideration. The corrections in the manuscript were marked by “Track Changes” function, and the responses to the comments are listed below. We hope the manuscript is now suitable for publication in Foods.

Sincerely yours,

Jie Zheng

Response to the comments

Since the article already has a summary, in my opinion, the authors should change the section "4. Summary" to Conclusions. Concerning this point, one of the reviewers suggested changing conclusions to "summary", I suppose because the information provided is a summary of the paper.

However, I think it is better to put back Conclusions and change the wording of the section to present to the readers the conclusions drawn from the study.

Response: Thank you for the suggestion. We have changed it to “Conclusion” and modified the description in this section.